# Preparation of High Molecular Weight Poly(urethane-urea)s Bearing Deactivated Diamines

**DOI:** 10.3390/polym13121914

**Published:** 2021-06-09

**Authors:** Alejandra Rubio Hernández-Sampelayo, Rodrigo Navarro, Ángel Marcos-Fernández

**Affiliations:** 1Institute of Polymer Science and Technology (ICTP-CSIC), Juan de la Cierva 3, 28006 Madrid, Spain; alerubioh@hotmail.com (A.R.H.-S.); amarcos@ictp.csic.es (Á.M.-F.); 2Escuela Internacional de Doctorado de la UNED, Universidad Nacional de Educación a Distancia (UNED), C/Bravo Murillo, 38, 28015 Madrid, Spain; 3Interdisciplinary Platform for “Sustainable Plastics towards a Circular Economy” (SUSPLAST-CSIC, Madrid, Spain

**Keywords:** high performance polyurethanes, deactivated diamines, silylation protocol, high molecular weight, DFT framework, condensed Fukui functions

## Abstract

The synthesis of poly(urethane-urea) (PUUs) bearing deactivated diamines within the backbone polymer chain is presented. Several deactivated diamines present interesting properties for several applications in the biomaterial field due to their attractive biocompatibility. Through an activation with Chloro-(trimethyl)silane (Cl-TMS) during the polymerization reaction, the reactivity of these diamines against diisocyanates was triggered, leading to PUUs with high performance. Indeed, through this activation protocol, the obtained molecular weights and mechanical features increased considerably respect to PUUs prepared following the standard conditions. In addition, to demonstrate the feasibility and versatility of this synthetic approach, diisocyanate with different reactivity were also addressed. The experimental work is supported by calculations of the electronic parameters of diisocyanate and diamines, using quantum mechanical methods.

## 1. Introduction

The poly(urethane urea) (PUUs) polymers are a subclass of the generic polyurethane (PU) family that has been widely studied. These polymers are usually prepared by a polyaddition reaction of diols and diamines to diisocyanates in bulk or in polar aprotic solvents such as *N*,*N*′-dimethylacetamide (DMA) at different temperatures. When difunctional reactants are used, the resulting polymer is linear and thermoplastic. These polymers are prepared by reaction of a relatively long and flexible macrodiol with a diisocyanate and a relatively short diol (PUs) or diamine (PUUs) referred as chain extender. They can be considered as segmented block copolymers. The macrodiol segments constitute the so-called soft segments (SS), and the segments produced by the reaction of the diisocyanate with the chain extender the so-called hard segments (HS). In certain conditions, these segments can thermodynamically phase-separate, with the HS acting as physical crosslinks that determine the mechanical properties. The polymerization can be carried out by mixing all the reactants at once, one shot method or in two stages, where initially the diisocyanate reacts with the macrodiol yielding an end-capped isocyanate prepolymer which reacts, in a second step, with the chain extender (diamine in PUUs) giving the corresponding polymeric chain, prepolymer method [1,2,3]. This method is preferred in PUUs to the direct polycondensation of a mixture of diamines and diols with the diisocyanate compound because due to the high reactivity of amines compared to alcohols [4,5,6,7], in the early stages of the reaction, polymer chains rich in urea moieties will be preferentially formed. Subsequently, these urea chains could precipitate, leading to a stoichiometric imbalance and a composition drift [8]. Consequently, the growth of the polymer chain would be hampered and the materials obtained would reach limited mechanical performance [9]. The reaction of isocyanate and hydroxyl groups is usually catalyzed by tertiary amines and organometallic compounds whereas the reaction of isocyanates and amines usually do not need catalyst [4,10].

In the biomedical sector, PU and PUUs are used in a variety of biomedical applications, most prominently as blood sacs in ventricular assist or cardiovascular devices [11,12,13,14] and tissue engineering [15]. The interest for this type of promising elastomers resides in the ability to modulate the performance of these materials according to need. These materials are solution-processable elastomers that exhibit good mechanical properties [16], versatile processability, tunable properties, while simultaneously exhibiting good biocompatibility [17]. Due to the stronger hydrogen bonds formed by the urea groups in PUUs compared to urethane groups in PUs, usually a better phase separated morphology is achieved leading to better mechanical properties. For the synthesis of these PUUs, aliphatic and aromatic diamines are widely used as chain extenders [18]. When aromatic rings are introduced on the polymer structure, thermodynamic incompatibility is increased between soft segments and hard segments improving the phase separated morphology with strong influence on the mechanical properties. For biological applications, if the polymer must be biodegradable, it is usual to draw upon the use of aliphatic diamines since they are less toxic than their aromatic counterparts [19]. However, there are aromatic diamines that lack side-effects and provide great interest due to their high performance at a competitive price. In Figure 1, two attractive aromatic diamines are depicted.

The interest for diamine PABA lies in that its chemical structure is based on p-aminobenzoic acid. In fact, its structure is quite similar to the local anesthetic, benzocaine. The aminoesters derived from this acid are rapidly metabolized in the plasma, by the action of an enzyme, leading to metabolites that are easily excreted through the urine [20]. The average life of these products are very short.

On the other hand, 4,4′-(diaminophenyl) sulfone (p-DDS) is employed as an antibacterial drug to treat Hansen disease patients or for the treatment of acne or dermatitis [21]. Currently this compound is marketed under the name of Dapsone^®^. Its action mechanism consists of the elimination of p-aminobenzoic acid, avoiding bacterial reproduction.

For both types of diamines, the presence of electron-withdrawing groups (COOR or SO2) into the aromatic ring produces a deep decrease in the nucleophilicity of the amino groups. Therefore, the reactivity of these deactivated products with electrophiles would be more hampered. In polymer chemistry, this decrease in reactivity leads to polymers with lower molecular weights and undesirable performances.

The transformation of amines into isocyanates [22], or the use of phenyl dichlorophosphite [23] or triphenyl phosphite [24], as coupling agents are some of the synthetic protocols developed for the insertion of aromatic diamines into polymeric matrices. However, these approaches are limited to strongly activated diamines, since they lack electron-withdrawing groups, such as benzidine, 4,4′-oxydianiline, o-tolidine. The high toxicity and possible cancerogenic character of these aromatic diamines has discarded their introduction into biomaterials [25]. Consequently, the most used diamines for these biosystems is reduced to aliphatic diamines, such as putrescine, cadaverine or 1,6-hexamethylenediamine.

Other protocols developed in this field are based on silicon chemistry. Indeed, the silylation reaction of reactants has been previously used for the preparation of condensation polymers, such as poly(ether ketone)s [26], polyamides [27,28] and polyimides [29,30]. Silylation of deactivated diamines supposes an intermediate synthetic step with the corresponding purification protocol [31], and in order to avoid it, a simpler in situ method has been described [27]. The preparation of high performance poly(urethane-urea)s bearing deactivated aromatic diamines by this method has not yet been addressed. Therefore, this communication focuses on the validation of the in situ silylation protocol for the synthesis of high performances PUUs using deactivated biocompatible aromatic diamines as chain extenders. The synthetized polymers were characterized and their achieved molecular weight related to the diamine reactivity and the effect of polycondensation agents. To shed light on these results, density functional theory (DFT) calculations was carried out, suggesting that the activation energy of the addition reaction is reduced by silylation of diamines. Given widespread urethane-containing materials, PUUs are of promising potential in biomedical applications due to their excellent mechanical performances, facile preparation and manipulation.

## 2. Materials and Methods

Chloro(trimethyl)silane (Cl-TMS), 4-Dimethylaminopiridine (DMAP), Isophorone diisocyanate (IPDI) and tin(II) 2-ethylhexanoate (Sn(Oct)_2_) were purchased from Sigma-Aldrich (Madrid, Spain) and used as received. Methylene diphenyl diisocyanate (MDI) was supplied by Lubrizol and sublimed just before use. The aliphatic isocyanate 1,6-hexamethylenediisocyanate (HDI) was purified by distillation under reduced pressure. 4,4′-Diaminodiphenyl sulfone (p-DDS) was purchased from Aldrich and recrystallized from ethanol.

Commercial polycaprolactone diol (PCL2054) Capa^®^ 2203A (Mn = 2054 g∙mol^−1^), kindly supplied by Perstorp (Warrington, UK), was dried in vacuum at 60 °C for 3 h and stored in vacuum before use. *N*,*N*-Dimethylacetamide (DMAc) was dried by distillation over commercial polymeric MDI [1] and Pyridine (Py) was dried by distillation over CaH_2_.

^1^H-NMR spectra were acquired on a Varian Unity Plus 400 instrument (Palo Alto, CA, USA) at room temperature, using deuterated chloroform and deuterated DMSO as solvents. All recorded spectra were referenced to the residual solvent signal 7.26 and 2.50 ppm for CDCl_3_ and DMSO-d_6_, respectively.

Fourier transform infrared (FTIR) measurements were carried out on a PerkinElmer Spectrum One spectrometer (Perkin-Elmer, Waltham, MA, USA) coupled with an attenuated total reflection (ATR) device. Sixteen scans were averaged from 4000 to 650 cm^−1^ and with a resolution of 2 cm^−1^.

The thermogravimetric analysis (TGA) of samples was carried out in a Mettler-Toledo TGA/SDTA 851 instrument (Mettler-Toledo, Schwerzenbach, Switzerland) from room temperature to 600 °C under a nitrogen atmosphere at a 10 °C/min heating rate.

The thermal transitions of the samples were analyzed by Differential Scanning Calorimetry (DSC) on a Mettler Toledo DSC 822e calorimeter (Schwerzenbach, Switzerland) equipped with a liquid nitrogen accessory. Discs cut from sheets weighing approximately 30–40 mg were sealed in aluminum pans. Samples were heated, from 25 to 90 °C at a rate of 10 °C∙min^−1^, cooled to −90 °C at the maximum rate of the instrument, maintained for 5 min at this temperature and re-heated from −90 to 200 °C at a rate of 10 °C∙min^−1^.

The mechanical properties were measured in an MTS Synergie 200 testing machine equipped with a 100N load cell. All the test specimens analyzed were cut with the dimensions established in standard ISO37 (type 4). A cross-head speed of 5 mm∙min^−1^ was used and the strain was measured from cross-head separation and referred to a 10 mm initial length. For all synthesized polymers, a minimum of 5 specimens were analyzed.

Gel permeation chromatography (GPC) analyses were carried out with Styragel (300 × 7.8 mm, 5 mm nominal particle size) Water columns. DMF with LiBr (0.1 *w*/*w*) was used as solvent. Measurements were performed at 70 °C at a flow rate of 0.7 mL∙min^−1^ using a RI detector. Molecular weights of polymers were referenced to PS standards.

### Poly(urethane-urea) Syntheses Following the Silylation Protocol

Poly(urethane-urea)s were prepared by reacting PCL2054 as macroglycol, with a diisocyanate to prepare a prepolymer that was subsequently reacted with a diamine. Thus, a PCL2054-PABA-HDI polymer will be composed of PCL2054 macroglycol, PABA diamine and HDI diisocyanate. In a typical run, 0.617 mmol of PCL2054, 3.110 mmol of IPDI and 2 drops of Sn(Oct)_2_ were dissolved in 2 mL of anhydrous DMAc. The reaction mixture was heated at 80 °C for 7 h. In another reactor 2.403 mmol of p-DDS were weighed and dissolved in 3 mL of anhydrous DMAc. The solution was cooled in an ice bath and 4.80 mmol of Cl-TMS was slowly added. After the addition, 4.800 mmol of anhydrous pyridine and 0.480 mmol of DMAP were added. This reaction mixture was kept stirring for further 15 min at 0 °C. Subsequently, the amine solution was incorporated into the first reaction mixture and kept stirring at room temperature for 24 h. At the end of this time, the polymer was precipitated on a water bath and thoroughly washed with water to eliminate all water-soluble bases. The polymer was finally recovered by filtration and dried overnight under vacuum. Films of the polymers were obtained by casting from DMAc solution (previously heated at 80 °C when the polymer did not dissolve at ambient temperature) into a Teflon sheet covered by a conical funnel and the solvent evaporated at 80 °C for 8 h. Finally, the film was dried in high vacuum at ambient temperature for 24 h.

## 3. Results

### 3.1. Model Reaction

Initially, the study was focused on the synthesis of model polyureas, since they act as a hard phase within the matrix of segmented poly(urethane-urea)s. Ensuring that the diamine activation protocol leads to soluble polymers during polymer synthesis is a key element in obtaining high molecular weight polymers because precipitation, stoichiometric imbalance, composition drift issues are completely avoided. Firstly, the silylation of the aromatic diamines followed the experimental conditions described by Lozano et al. [29]. Freshly-distilled Chloro-trimethylsilane (Cl-TMS) was used as a silylating agent and a base set of Pyridine (Py) and (4-Dimethylaminopyridine) DMPA, in a 1:0.1 mol ratio, as catalysts. (Scheme 1).

Initially, deactivated diamine was dissolved with pyridine and DMAP in anhydrous DMAc, then the silylating agent (Cl-TMS) was added dropwise to avoid undesired increases in reaction temperature. Indeed, the silylation reactions were carried out at 0 °C. The activation reaction of diamine compounds by silylation proceeded homogeneously, under these experimental conditions, and precipitates were not detected. Then, in the second step of this protocol, the in situ silylated diamines reacted with diisocyanate products yielding the desired polyureas in a high yield.

It should be highlighted that the order of addition of the reactants is a determining factor since it establishes a correct progress of the reaction. Occasionally, silylated products may be accompanied by precipitation if the addition reaction to electrophiles does not occur properly, leading to lower yields and molecular weights [23].

The synthesis of polyureas through a silylation reaction of amines was carried out by Prof. Dr. Imai et al. in 1992 [32]. Nevertheless, the authors presented a series of significant differences with respect to our current work. Firstly, the authors described the protocol for the preparation and isolation of the silylated diamines. Thus, these silylated diamines were difficult to handle and to isolate, and required very rigorous anhydrous conditions to avoid side-reactions. To overcome this issue in our protocol, an in situ silylation method has been proposed carrying out the polyaddition reaction in a single step. In this way, employing an excess of Cl-TMS, traces of water presented in the reaction medium could be easily removed and the side-reactions could be completely inhibited. Secondly, the condensation between silylated amines and diisocyanates was carried out at higher temperatures, between 40 to 100 °C and they also employed fewer usual solvents for polyaddition reactions such as Toluene, THF or DMSO. Except for DMSO, the other solvents might have some difficulties in dissolving high molecular weight polymers due to their low polarity, hence with these solvents, the degree of polymerization could be negatively compromised. Eventually, the hygroscopic nature of DMSO could adversely affect the progress of the polyaddition reaction.

In our case, the polyaddition reaction occurs at much lower temperature (0 °C) taking advantage of the high reactivity of silylated diamines and avoiding excessive heating that could lead to side-reactions. Moreover, we have also proposed aprotic solvents with high polarity, such as DMAc, DMF or NMP. These solvents are more usual for the preparation of polyureas.

Finally, and probably most importantly, the diamines used in that previous work lacked electron-withdrawing, that is, the diamines employed were indeed highly reactive due to their chemical structure. On the contrary, the essential motivation of our work is to introduce deactivated biocompatible diamines within the polymer chain though a simple activation protocol.

Table 1 shows the molecular weights of polyureas obtained for Diamine PABA with three types of isocyanates (aromatic, symmetric aliphatic and asymmetric aliphatic). As reference, the corresponding polyureas were prepared in parallel by the equimolecular reaction between diisocyanates and the diamine PABA in the absence of any type of silylating agent. For both aliphatic diisocyanates (HDI and IPDI), it is observed that the polyureas obtained by the silylation method of the diamines present higher molecular weights than standard protocol. This demonstrates that the activation of the diamines is an efficient process for obtaining polymers with higher molecular weights. The polyureas derived from the aromatic isocyanate (MDI) were only soluble in boiling DMAc, preventing the determination of their molecular weights by GPC. Since this aromatic isocyanate is highly symmetric and reactive, polyurea chains would form well-organized structures, which could hinder dissolution of these polyureas. Gel permeation chromatography (GPC) chromatograms of synthetized polyureas are presented in Appendix A.

For polyureas derived from aliphatic diisocyanates, with lower reactivity than aromatic diisocyanates, the influence of the activation of the diamines on the molecular weights became more remarkable. For instance, the reaction of asymmetric diisocyanate IPDI with PABA under standard conditions led to very low molecular weights, close to 10 kDa; however, the same reaction assisted by silylating agents led to polyureas with molecular weights two-fold higher. This shows how activation of the aromatic diamine is required to achieve good performance with this type of polymers. It is noteworthy that the polyureas derived from the methanolysis of the N-silylated polymers have low crystallinity and good solubility in organic solvents, in contrast to the polyureas prepared by the diamine-diisocyanate route. The ^1^H-NMR spectrum of the polyurea derived from PABA and HDI is shown in Figure 2.

Owing to the absence of signals around 0 ppm, it is concluded that the activating group (trimethylsilyl moiety) has been removed during polymer precipitation, as shown in Figure 2. Our finding is in agreement with that described by Imai et al. [31], that demonstrated that silylated diamines easily lose the activating group by methanolysis. In addition, there are no signals from the basic catalytic agents (pyridine or DMAP) used in the addition reaction.

### 3.2. Poly(urethane-urea) Polymer Synthesis

After establishing the reaction conditions for the preparation of polyureas (as a hard segment model) by activating the amino groups, this methodology was extrapolated for the synthesis of PUUs. In the chemical structure of these polymers, one can define a soft segment (polyester or polyether type) and a hard segment (polyurea type). This segmented structure leads to very interesting elastomeric properties for different applications. According to the weight ratio between both phases the obtained polyurethane presents different performances and features. The synthesis of these polymers was carried out in two steps, firstly a prepolymer was formed, which results from the reaction between the macrodiol and excess diisocyanate. Due to the excess of diisocyanate, end-capped diisocyanate oligomeric chains would be obtained. Later on, these oligomers would react, in a second step, with an equivalent amount of diamine for the growth of the polymer chains of PUUs.

The synthesis of the end-capped functionalized prepolymers was carried out in a concentrated solution of DMAc and using Sn(Oct)_2_ as catalyst. The reaction temperature was 80 °C, which is a usual temperature for the synthesis of polyurethanes. As shown in Figure 3, the reaction progress was followed by the variation of the isocyanate band (located at 2260 cm^−1^). All the ATR-FTIR spectra were normalized to the band at 1630 cm^−1^, because it remained constant during the reaction; this band was related to the carbonyl stretching vibration of DMAc. In all the tested cases, it was observed that the isocyanate band progressively decreased with reaction time, until reaching a constant value. When the isocyanate band remained constant, the complete prepolymer synthesis had been achieved. The band related to the urethane group appears as a shoulder to ester band (1730 cm^−1^), preventing the monitoring of this band to follow the prepolymer synthesis.

Depending on the nature of the diisocyanate, the reaction time for prepolymer formation considerably varied. Thus, when an aromatic diisocyanate was used, the initial reaction time was reduced to 2 h, however, employing aliphatic diisocyanates, the reaction time increased to 7 h. This time difference is related, again, with the high difference in reactivity between the diisocyanates used. The aromatic isocyanate derivative required shorter reaction times due to its higher reactivity [33]. Once the end-capped isocyanate prepolymer was formed, the activated diamines with silylating agents were added. The additions were carried out at low temperature (0 °C) to control the reaction and stirring was maintained overnight. In the absence of silylating agents, the polyaddition reaction of diamines on the prepolymer was carried out at higher temperatures, especially when the prepolymer came from the asymmetric aliphatic diisocyanate IPDI that has a lower reactivity than the symmetric aliphatic diisocyanate HDI. Finally, the PUUs were precipitated on excess of distilled water and the washing process was repeated several times, to remove all the bases and silylating agents. The final dry PUUs were analyzed by different techniques and their mechanical properties were measured.

### 3.3. Polymer Characterization

Table 2 shows the molecular weights of the synthetized PUUs. Except for the PCL2054-PABA-HDI system, all prepared PUUs were formulated to have a 50% by weight of hard segment (50% HS or 50 HS), defined as (weight of diisocyanate + weight of diamine) × 100/total weight. Attempts to increase the hard segment fraction for the PCL2054-PABA-HDI system were unsuccessful because the polymer precipitated during the reaction and prevented the increase in molecular weight. This anomalous behavior was detected for both the classical synthetic route and the silylation protocol. Therefore, it was necessary to reduce this content in the hard segment in order to evaluate the silylation protocol on the molecular weight for the PCL2054-PABA-HDI system.

Unlike what happened in the model polyureas, now the influence of the reactivity was more remarkable for aromatic (MDI) than for aliphatic diisocyanates (HDI, IPDI) where the increase in molecular weight of the final polymer increased slightly. This could be due to the fact than in the first stages of prepolymer formation, the growth of the soft segments is hindered by the chemical structure of IPDI. As a consequence, in the final polymer, phase separation between the soft and hard segments is more impeded, leading to lower performance and mechanical properties. On the other hand, the most significant change was detected for silylated PUUs based on MDI, because they were only completely soluble in hot boiling solvent (DMAc). This finding is in line with the results described above for polyureas. Despite the introduction of a soft segment, only the non-activated MDI-based PUUs were soluble at room temperature and their molecular weights were determined by GPC.

The chemical structure of all the synthesized polymers was confirmed by NMR. In Appendix A, the chemical structure of the synthesized poly(urethane-urea)s is presented with the assignments of the proton NMR peaks. Carbon NMR spectra are also included in the Appendix A and the detailed characterization of the polymers including the assignation of the carbon NMR peaks will be the subject of a future work. As seen in Appendix A, the proton NMR peaks for the PUUs prepared in standard conditions and with the silylation protocol are the same proving that the chemical structure is identical and that during the precipitation of the polymer in water the activating silylating group is completely removed.

When the thermal properties are compared, the degradation of the silylated PUUs takes place at higher temperatures, as seen in Appendix A. It is known that the urethane and urea groups are the first groups to degrade in segmented polyurethane and poly(urethane-urea)s [1], thus the longer hard segments produced in the silylated PUUs is the reason for the increase in the temperature of the peak in the derivative (temperature for the maximum on the decomposition rate) of the TGA curve. Since the chemical structure is identical, no significant differences are expected for the thermal properties between the materials with or without silylation. As an example, sample PCL2054-PABA-HDI with 30% weight of hard segment was examined. As seen in Appendix A in the second heating, a low temperature Tg is observed due to the phase separated polycaprolactone soft segment, followed by very small melting endotherms due to soft segment melting and hard segment melting. Both curves are very similar showing no significant differences between both materials.

The results of the mechanical properties are collected in Table 3. For all tested cases, the mechanical properties of the silylated PUUs were generally better than the PUUs obtained under standard conditions. Therefore, through this protocol, poly(urethane-urea)s bearing deactivated diamines can be obtained with attractive mechanical properties. The stress/strain curves for each pair of PUUs are shown in Appendix A. Silylation improved the mechanical properties of polymers showing the critical effect of the polymer molecular weight on the mechanical properties.

### 3.4. Theoretical Study

To demonstrate that the silylation reaction of the diamines allows obtaining PUUs with better performance, a series of computational studies were carried out within the DFT framework. It is well-known that in the addition reaction of an amine to an isocyanate, the nucleophilic character of the amine is related to the energy gap between the highest occupied molecular orbital (HOMO) of the amine and the lowest unoccupied molecular orbital (LUMO) of the isocyanate. Therefore, the lower the difference is between both orbitals, the greater the amine reactivity becomes and thus the greater the possibility is of forming the urea group. During the addition reaction between an aromatic amine and an isocyanate in the presence of TMS-Cl, two steps can be distinguished. Firstly, the silylation of amine and secondly the addition of the silylated amine to an isocyanate.

It has been proposed [30] that the silylation of amines is catalyzed by the presence of a base, such as tertiary amine (Pyridine or DMAP). However, it is necessary to control the basicity of the base catalyst since if its basicity is very high the diamine could be disilylated, yielding to a less reactivity toward electrophiles because of steric effects.

The improvements achieved in the performance and mechanical properties of the silylated PUUs were strongly reflected when the aromatic diisocyanate MDI was used (Efs 1 and 7), allowing to considerably exceed the modulus and maximum stress with respect to the PUUs synthesized by the classical protocol (Entries 2 and 8). It is worth noting the increase of modulus value of polymer based on IPDI after silylation, which increased 2-fold. The optimized structures of pristine, silylated diamines and diisocyanates are collected in the Appendix A). From these optimized structures, the energy of the frontiers’ orbitals (HOMO and LUMO) can be established, which are responsible for the chemical behavior shown by these molecules.

The data obtained for the E_HOMO_ of pristine and silylated diamines and E_LUMO_ of diisocyanate are shown in Table 4. The introduction of trimethylsilyl moiety into the amine group leads to an increase the energy of the HOMO orbital, making it easier for the addition to an isocyanate group to form a urea group.

According to Table 4 for both types of amines (PABA, p-DDS) the energy gap between silylated amines and electrophiles (isocyanates) decreases respect to unsilylated amines. Therefore, the silylation reaction should decrease the activation energy of addition reaction, considering that this energy gap is responsible of the chemical reactivity. This consideration corresponds when aromatic diisocyanate MDI is used, however in aliphatic systems this consideration fails. Based on frontier molecular orbital theory, IPDI should be more reactive than HDI, however, there are additional factors such as steric hindrance that hampers the reactivity of IPDI. Thus, experimentally it is observed that HDI is more reactive than its partner IPDI. In the same way, in principle the DDS amine should be more reactive than PABA, however, the experimental results show the opposite tendency. To explore this discrepancy, it is necessary to calculate the local reactivity descriptors (such as Fukui functions). These functions predict the reactivity of atoms in a molecule against nucleophiles and electrophiles. The condensed Fukui function values were obtained by UCA-Fukui software 1.0 [34].

Condensed Fukui functions values for both nitrogen atom of the amines are collected in Table 5. In this table, it is concluded that the p-DDS possesses both nitrogens with equally reactivity towards electrophile, whereas in the PABA its chemical reactivity is strongly focused on an amino group. This difference in reactivity would be associated to the fact that the p-DDS has an extended conjugation between both aromatic rings whereas PABA this conjugation is avoided due to the presence of a linker between both aromatic rings (trimethylene group).

Indeed, the results obtained show that almost 35% of the reactivity of the diamine PABA is focused on a single atom while in the p-DDS this reactivity is distributed between both nitrogen atoms. It should also be noted how the value of condensed Fukui function rises up to 40% in both cases, due to the insertion of the silyl moiety. However, despite the increase, the function remains concentrated in a nitrogen atom in PABA and evenly distributed in p-DDS.

## 4. Conclusions

The introduction in situ of silyl groups in the amino groups of deactivated diamines has proved to be an efficient method to obtain PUUs with high molecular weight and better performances.

The silylation was carried out in the reaction medium without isolating the corresponding silylated diamines, simplifying the handling of these type of activated diamines and avoiding side-reaction effects. Proton NMR proved that the chemical structure of the PUUs without and with silylation activation was identical after removal of the silylation agent by precipitation in water. The higher molecular weight obtained by using the silylation protocol improved significantly the thermal stability and the mechanical properties of the synthesized PUUs.

To corroborate the reactivity observed experimentally, theoretical studies were carried out which demonstrated that silylation decreased the activation energy of the addition reaction. The increase in energy of the HOMO orbital after the insertion of the silyl group is responsible for the decrease in the activation energy.

This work shows the feasibility of in situ silylation methodology for the insertion of deactivated diamines within the PUUs chains, and for obtaining polymers with good performance.

## Data Availability

The data presented in this study are available on request from the corresponding author.

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
