# Peer review of "Preparation of High Molecular Weight Poly(urethane-urea)s Bearing Deactivated Diamines"

_polymers, 2021, doi:10.3390/polym13121914_

Round 1
Reviewer 1 Report
After a careful check of the received publication, I cannot accept publishing this work in its current version. This work can be published only if the revised version will carefully address ALL required points (below). Major changes must be addressed before further consideration.
This work lacks novelty that can attract readers of Polymers, and some important characterizations such as DSC to investigate the thermal properties of the resulting polymers and to know if they are amorphous, crystalline, semi-crystalline polymers, addition of a detailed discussion of NMR section including carbon peaks assignments to confirm the chemical structures of the prepared materials (where authors added 13C NMR spectra in SI file without any peak assignment and interpretation in the main text..), thereby proving the success of the aimed method. I also wonder why the authors did not add in the main text a relevant paragraph reporting the most representative degradation parameters of the polymers obtained from TGA analysis, such as the weight of residues at 500°C (R 500 °C ), maximum decomposition temperature (Td,max), and the degradation temperatures at 5% and 10% weight loss (Td,5% , Td,10%). They only added the TGA thermograms in SI file without any relevant paragraph…
I found an obvious contradiction in this study; the authors reported the concept of biocompatibilty of the diamines used here as well as their advantages, etc. But, in reality, the MDI and HDI diisocyanates used here to prepare PUUs are well-known as very toxic diisocyanates and they show very dangerous effects (mutagenicity, carcinogenicity, reproductive toxicity, etc.). In addition, the EU recently adopted restrictions on the use of MDI and HDI…
-Authors should reformulate the introduction in order to obviously highlight the main aim of this study. This section should be much improved and well structured.
-Section 2.1, line 122, page 3 (Polyurethane-urea syntheses following the silylation protocol):
It is written “In a typical run, 0.617 mmol of PCL-2054, 3.110 mmol of IPDI and 2 drops of Sn(Oct)2…”,
-Page 7, Fig. 3, line 246: it is written “ATR-FTIR spectra of prepolymer synthesis of PCL-2054 and IPDI catalyzed by Sn(Oct)2”
-In Supplementary Materials: it is written “PABA-MDI-PCL2054-50HS / PABA-HDI-PCL2054-30HS / PABA-IPDI-PCL2054-50HS / p-DDS-MDI-PCL2054-50HS / p-DDS-MDI-PCl2054-50HS, etc.”
I wonder what does it mean by PCL2054 attached always to polymer sample name? As well as the meaning of 30HS or 50HS? I want to know the role of PCL2054 in drying procedure of DMAc. Relevant references reporting the use of PCL-2054 for such purpose..
-Table 2 and Table 3 in the main text: What does it mean by “HS (%wt)”
- what is the difference in terms of data provided (GPC) in Table 1 and Table 2 for the same polymers: for example in table 1, for PABA/HDI with silylation, MW is found to be 25.7 KDa and dispersity is 1.49. In table 2, for the same polymer (PABA/HDI with silylation), MW is found to be 113 KDa and dispersity is 1.9??
- The authors must explain in detail and in a distinct manner the difference in terms of the thermal behavior (DSC) of the resulting materials prepared with and without silylation, as well as in terms of the microstructes that could lead to a difference in properties, thereby highlighting the benefits of the silylation strategy exploited herein…
Author Response
Attached file
Reviewer 2 Report
The authors have revised the article according to the suggestions, I think that the article could be accepted at the present form.
Author Response
This is the manuscript highlighting the changes

Reviewer 3 Report
The article was prepared correctly.
The topic related to the search for new substrates, including extenders for the production of polyurethanes with special properties, is extremely interesting for scientists, but also for producers of these materials. It is also interesting that the authors linked this search with the use of polyurethanes in medicine.
Author Response
This is the final manuscript without highlighting changes

Round 2
Reviewer 1 Report
After a careful reading of the revised manuscript, the latter is accepted for publication in Polymers in its present form.